# Mesenchymal Stem Cells: The Secret Children's Weapons against the SARS-CoV-2 Lethal Infection

**Mario Giosuè Balzanelli** [1], **Pietro Distratis** [1], **Orazio Catucci** [1], **Angelo Cefalo** [1], **Rita Lazzaro** [1], **Francesco Inchingolo** [2], **Diego Tomassone** [3], **Sergey K. Aityan** [4], **Andrea Ballini** [5,6,*], **Kieu C.D. Nguyen** [7] and **Ciro Gargiulo Isacco** [1,2]

1. SET-118, Department of Pre-Hospital and Emergency-San Giuseppe Moscati Hospital, 74100 Taranto, Italy; mario.balzanelli@gmail.com (M.G.B.); distratispietro@gmail.com (P.D.); oraziocatucci@live.it (O.C.); ugemel@yahoo.it (A.C.); rita-lazzaro@libero.it (R.L.); drciroisacco@gmail.com (C.G.I.)
2. Department of Interdisciplinary Medicine, University of Bari "Aldo Moro", 70124 Bari, Italy; francesco.inchingolo@uniba.it
3. Foundation of Physics Research Center, 87100 Cosenza, Italy; dietomoh@gmail.com
4. Director of Multidisciplinary Research Center, Lincoln University, Oakland, CA 94612, USA; aityan@lincolnuca.edu
5. Department of Biosciences, Biotechnologies and Biopharmaceutics, University of Bari "Aldo Moro", Campus Universitario "E. Quagliariello", 70125 Bari, Italy
6. Department of Precision Medicine, University of Campania "Luigi Vanvitelli", 80138 Naples, Italy
7. American Stem Cells Hospital and Human Stem Cells Institute, Ho Chi Minh City 70000, Vietnam; drkieukaren@gmail.com
* Correspondence: andrea.ballini@uniba.it

**Abstract:** Due to the promising effects of mesenchymal stem cells (MSCs) in the treatment of various diseases, this commentary aimed to focus on the auxiliary role of MSCs to reduce inflammatory processes of acute respiratory infections caused by the 2019 novel coronavirus (COVID-19). Since early in 2020, COVID-19, a consequence of the severe acute respiratory syndrome coronavirus 2 (SARS-CoV-2), has rapidly affected millions of people world-wide. The SARS-CoV-2 infection in children appears to be an unusual event. Despite the high number of affected adult and elderly, children and adolescents remained low in amounts, and marginally touched. Based on the promising role of cell therapy and regenerative medicine approaches in the treatment of several life-threatening diseases, it seems that applying MSCs cell-based approaches can also be a hopeful strategy for improving subjects with severe acute respiratory infections caused by COVID-19.

**Keywords:** COVID-19; SARS-CoV-2; mesenchymal stem cells (MSCs); induced pluripotent stem cells (iPSCs); embryonic stem cells (ESCs); immune system

## 1. Introduction

The adaptive immunity plays a crucial role in SARS-CoV-2 infection: In fact, proinflammatory mediators activate either the type 1 T helper cells (Th1) immune response (CD4+ and CD8+ T cells) or B lymphocytes that trigger the virus-specific immune response [1]. However, the clinical scenario of adults infected by SARS-CoV-2 usually reveals a marked lymphocytopenia, together with a lower B lymphocyte count [2].

The percentage of severe COVID-19 infected adults is incomparable to that of young individuals. Of note, healthy pediatric subjects have revealed a higher count in total lymphocytes compared to healthy adults. In addition, children infected with SARS-CoV-2, in accordance with data of Giuseppe Moscati Hospital (Taranto, Italy), showed a peripheral blood lymphocytes normal count, indicative of lesser immune stress [3]. Since, children exhibit a higher and more frequent exposure to external pathogens this would certainly contribute to a higher need in immunity expression thus, the presence of mild disease may

refer to a "trained immunity" exerted by innate immunity cells known as "memory cells" after the antigen exposure to pathogens [4–6].

These systemic antigens determine a series of functional changes either transcriptomic or metabolomic within bone marrow and peripheral blood. These changes activate and enhance the expression of haemopoietic progenitors and mesenchymal stem cells (MSCs), together with Natural Killer (NK) and innate lymphoid cells leading to the formation of a group of immune cells equipped with stronger and faster defensive system against infections [4]. Recent findings showed that the entire repairing and re-organizing mechanism takes more time and often it does not happen at all. SARS-CoV-2 enters the lungs via angiotensin-converting enzyme 2 (ACE2) receptors, and induces an immune response with the accumulation of immunocompetent cells, causing a cytokine storm, which leads to target organ injury and subsequent dysfunction [7].

For instance, the lung lymphoid cells were shown to remember their status once triggered by previously inhaled specific allergens, similarly to the NK cells that can recognize particular pathogens, such as the cytomegalovirus and flu A virus, starting a mediated secondary innate immune response as soon as the they re-encounter the same intruder [7]. It is demonstrated that also common epigenetic mechanisms determine memory cell development, both in the adaptive and innate immune system [5].

In pediatric subjects, the frequent respiratory infections potentially damage the inner mucosa, tissues, and epithelial cells of the lung constantly triggering the surveillance presence of immune cells. In this scenario, local MSCs were shown to interact with local immune system inducing the activation of specific immune cells such as gamma and delta T lymphocytes ($\gamma/\delta$ T Cells), macrophages type 2 (M2). In addition, MSCs almost immediately are seen to differentiate into alveolar type II (ATII) epithelial cells in order to start the repairing mechanism, these cells form approximately 60% of the pulmonary alveolar epithelium [1,6].

In young subjects all these events rarely take place and it has probably to be linked to the regenerative patterns typical of this age. The higher presence of MSCs generate an immediate response to inhibit the production of pro-inflammatory cytokines and interleukins such as TNF-a, IFN-y, IL-6, IL-2, and IL-1. The anti-inflammatory activities of MSCs take place in two complementary steps, first influencing the monocyte differentiation towards dendritic cells (DCs) and M2 and, second by inducing tolerant phenotypes of naive and effector T cells and, inhibiting antibody production by B cells together with a suppressing activity on NK cell proliferation and NK cell-mediated cytotoxicity [8,9] (Figure 1).

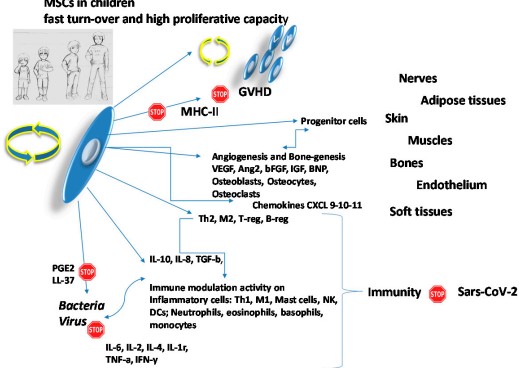

**Figure 1.** MSCs perform different activity. In young subjects these cells showed high rate of proliferation and turn-over. The MSCs in children, teenagers, and young adults are able to keep their status in differentiating state and secrete factors that can interact at different cellular processes, such as immunomodulation, chemoattraction, progenitor cell proliferation, angiogenesis, skeletal, and muscle genesis. Their fast differentiation capacity, their long telomeres keep the ACE2 receptors in uncompleted condition which does not allow the SARS-CoV-2 entrance, furthermore the high proliferation grade and their immune modulation ability may have an inhibitory effect on virus invasion ability.

## 2. Hypothesis: Why Stem Cells Could Be a Possible Solution

Given the special immunomodulatory property of MSCs, together with their low major histocompatibility class I (MHC-I) expression, MSCs showed to be able to treat in vivo the steroid-resistant graft-versus-host disease (GVHD) in patients underwent allogeneic hematopoietic stem cells and solid organ transplantation improving the outcomes of clinical diseases as a consequence of aberrant and destructive immune responses. In addition, the MSCs high number and their fast turn-over particularly seen in young subjects may contribute to clarify their immunity to SARS-CoV-2. The idea behind this assertion is also based upon the fact that the SARS-CoV-2 fast tropism would be eventually its greatest weak point, as it tends to lose efficacy each time has to replicate following the high presence of a strong immune modulatory environment [9–11].

The diminished capacity of adults and elderly to effectively respond to the threat of SARS-CoV-2 may thus reside in the limited self-renewing capacity of circulatory MSCs [12,13]. During every stem cell division there is a price to pay, which is the shortening of each single cell telomere [14]. To better understand the importance of this matter it is necessary to clarify the essential role of telomere in maintaining the full cell integrity which, in turn, means healthy tissues, organs, and active regenerative functions [12]. Telomeres are specialized DNA-protein arrangements that firmly sealed the ends of linear chromosomes. The telomeres need constant extension of telomeric DNA repeats to preserve chromosomes stability and conserved [15]. Therefore, anything that may negatively affect telomeres may eventually drive to chromosome dramatic changes driving to harsh systemic consequences [12].

In addition, the telomere shortening is amplified during chronic and acute inflammation and oxidative stress, as these condition stimulate cell division necessary for tissue repair and immunology responses. A clear examples came from study that showed how shortened leucocyte telomeres, has to be considered a marker of immunosenescence, indicative of weakened replicative capacity that highly contribute to lower resistance towards infections, condition often seen in aged patients or chronic ill individuals. Therefore, following the above observations, it should not be a surprise that most deaths caused by COVID-19 occur among frail chronic ill individuals with preexisting comorbidities [12]. Lymphocytopenia, characterized by a low number of T helper cells CD4+, CD8+, and B lymphocytes, is highly suggestive of poor prognosis in adult COVID-19 infected patients indicative of systemic failure and defected immunity [9,10].

Further, the results obtained from different studies clearly demonstrated that cells of tissues with high turn-over such as skin and bones go through a quicker telomere shortening which indicates the steady progression of the aging process confirmed by the increase in wrinkles, skin maculation, and bone and joint diseases [12]. Whether telomere is directly responsible to the insurgence of a disease or it is just a consequence of it, this is still a matter of debate and conclusion still controversial.

Scientific findings have confirmed that the regenerative capacity of the body could be potentially a process that goes throughout the adult life almost independently from the individual's age. The mechanism was finely elucidated by Shimamoto et al. in regarding the induced pluripotent stem cells (iPSCs) [12]. Their assumption to obtain pluripotent stem cells from normal adult cells was eventually confirmed by molecular engineering through the insertion into the cells of genes such as Oct4, Sox2, Nanog, Osteocalcin (OCN), Nestin in charge of pluripotency and multipotency. Of note, these are the same genes that were characterized by our team on circulatory stem cells collected from human peripheral blood [13–15].

Assuming that the relationship between the length of telomere and aging is a multifactorial mechanism [13], the ongoing regenerative process in children may represent the key factor that explains the immunity to COVID-19 exposures (Figure 2).

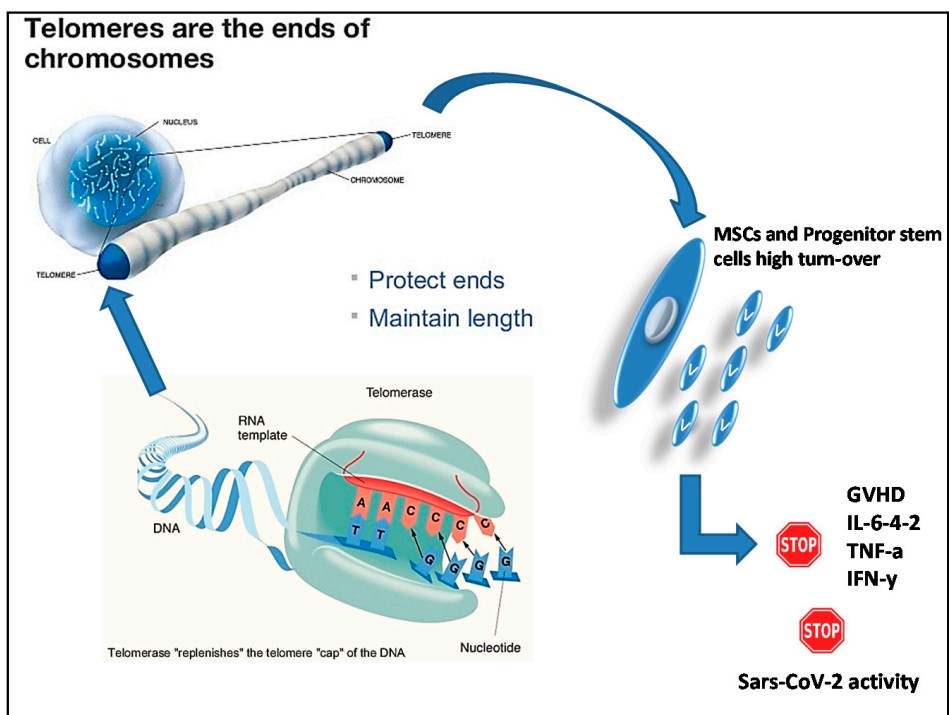

**Figure 2.** Hypothesis linking telomere length and replicative capacity of MSCs to autoimmune pathophysiology in contrasting SARS-CoV-2 aggression in young adults and children. The image indicates how telomere is a key factor in sustaining the replicative capacity of individual MSCs population doublings which then activate their immune modulation property against the inflammatory process lead by IL-6, TNF-$\alpha$, IFN-$\gamma$ triggered by the virus.

### 3. Present Procedure and Future Perspectives and Limitations

In the era of regenerative medicine, the MSCs have been considered as a novel treatment procedure for many infectious diseases such as HBV, HCV, and HIV due to their ability to home to damaged tissues, hypo-immunogenicity that allows allogenic transplantation, anti-inflammatory effects, and their differentiation capacity into functional mature cells such as hepatocytes, osteoblasts, and immune regulatory cells [7,16,17].

Thus, we can assert with a confident degree of certainty that high stability ACE2 receptor concentrations, trained immunity, and a constitutional high lymphocyte counted in children may only partially explain their capacity to be virtually immune to SARS-CoV-2 invasion and aggressiveness. Therefore, we speculated the presence of two key elements which contribute to the immunity of the young: first, the high number and fast turn-over of circulatory MSCs and progenitor stem cells; and, second, the long telomere that allow the stem cell fast replication, high numbers, and quick expansion. These findings may therefore contribute to explore an alternative way to treat the SARS-CoV-2 in adults, in elderly and compromised individuals with preexisting comorbidities and viral infection such as HIV [18–20].

Though there a differences between SARS-CoV-2, HIV, HCV, or HBV in the viral assembly and budding mechanism, these viruses have been more diverse and unpredictable in pathophysiology, sharing a similar invasive approach. For instance, the key host receptors used by SARS-CoV-2 surface structural spike glycoprotein (Sp) involves the ACE2 receptors, and the viral RNA machinery during the whole replicative process within a cell seems to be also dependent of CD4 receptors in a similar manner of HIV [20,21]. Engineered sequence performed on Sp in positions 21,840 to 21,855 of the COVID-19 sequence MN908947, showed a 15-bases match to the envelope glycoproteins (Envs) of the HIV-1 [19]. According to Luc Montagnier study group, the significant engineered sequences, are all crucial for the penetration into the cells of the human lungs [22].

So, it would be legitimate to assume that the use of MSCs might contribute not only to reduce the invasiveness and the aggressiveness of SARS-CoV-2, but should be also considered a valid tool in immunocompromised COVID-19 patients, as they would preclude any viral changes, which implies the formation of a reservoir of unresponsive molecules and factors towards endogenous immune inhibitors [20,23–28].

On the other hand, this hypothesis has some limitations due to the novelty of SARS-CoV-2, and consequently, the limited literature in this assumption. Besides, for future studies, both the use of MSCs clinical use and the in vivo analysis of telomere length in COVID patients could be challenging and carry a high risk for examiners.

### 4. Conclusions

Overall, two hypotheses have been assumed in this commentary. First, the integrity presence of a stable regenerative state of pediatric and young individuals, mainly based on the support of active circulating stem cells; and, secondly, the active presence of the adaptive immune system characterized by a continuous turn-over of lymphocytes, NK and B lymphocytes. We are well aware of the limitation of these scientific assertions and the we know that more evidence and studies need to be done to achieve more stable and constructive conclusions.

**Author Contributions:** All authors equally contributed to the manuscript. All authors have read and agreed to the published version of the manuscript.

**Funding:** This research received no external funding.

**Conflicts of Interest:** The authors declare no conflict of interest.

### Abbreviations

| | |
|---|---|
| ATII | Alveolar Type II |
| DCs | Dendritic Cells |
| ESCs | Embryonic Stem Cells |
| GVHD | Graft-Versus-Host Disease |
| iPSCs | Induced Pluripotent Stem Cells |
| MHC I | Co-Stimulatory Factors CD-80 and CD-86 |
| MSCs | Mesenchymal Stem Cells |
| M2 | Macrophages Type 2 |
| NK | Natural Killer |
| SARS-CoV-2 | Severe Acute Respiratory Syndrome Coronavirus 2 |
| $\gamma/\delta$ T Cells | Gamma and Delta T Lymphocytes |

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
