# Peer review of "Mesenchymal Stem Cells: The Secret Children’s Weapons against the SARS-CoV-2 Lethal Infection"

_applsci, doi:10.3390/app11041696_

Round 1

Reviewer 1 Report

The Authors introduced hypothesis that telomere length of MSCs and higher activity of adaptive immune system have influence on MSCs self-renewal and protect children and young  adults from SARS-Cov-2 infection. This is in line with current knowledge on MSCs biological properties showing that in aged organism a regenerative processes are slower. However, it is worth to add an information on lung stem/progenitor cells turnover in tissue homeostasis maintenance. The lung is a conditionally renewing organ, and in adult and elderly, the turnover of airway epithelial cells is less than 1% per day, in contrast to other organs eg. skin, as is discuss in the review paper [PMID: 33033561].

The sentence between lines 78-86  “Several are the evidences confirmed is ……” is too long, confusing and unclear. In this sentence is the phrase “…by acting on the differentiation of monocytes into T- lymphocytes, dendritic cells (DCs)…” which is misleading. In the proper conditions monocytes may differentiate into dendritic cells but not into T-lymphocytes. MSCs may interplay with T-lymphocytes modulating their function. This part of introduction need to be rephrased accordingly.

Lines 114-116  “…to treat in vivo the steroid-resistant graft-versus-host disease (GVHD) in patients underwent either allogeneic organ or hematopoietic stem cells transplantation….”. The GvHD is a systemic disease occurred after hematopoietic stem cells transplantation not after organ transplantation. This statement should be rephrased accordingly.

This manuscript need professional English editing.

Author Response

We thank the Reviewer for the considerable attention and the valuable comments that certainly helped us to improve the quality of the present paper. We have revised the manuscript according to the Reviewers’ comments. A revision of the article has been carried out (underlined in yellow for Referee 1 and green for Referee 2, in the main manuscript text).

Please let us know if the revised paper satisfies requirements for publication.

Thank you very much for your attention and courtesy.

Reply to Referee  report I (yellow in the text):

The Authors introduced hypothesis that telomere length of MSCs and higher activity of adaptive immune system have influence on MSCs self-renewal and protect children and young  adults from SARS-Cov-2 infection. This is in line with current knowledge on MSCs biological properties showing that in aged organism a regenerative processes are slower. However, it is worth to add an information on lung stem/progenitor cells turnover in tissue homeostasis maintenance. The lung is a conditionally renewing organ, and in adult and elderly, the turnover of airway epithelial cells is less than 1% per day, in contrast to other organs eg. skin, as is discuss in the review paper [PMID: 33033561].

R: We have really appreciated the kind suggestions from the reviewer and the paper was improved also in the reference. Thank you.

The sentence between lines 78-86  “Several are the evidences confirmed is ……” is too long, confusing and unclear. In this sentence is the phrase “…by acting on the differentiation of monocytes into T- lymphocytes, dendritic cells (DCs)…” which is misleading. In the proper conditions monocytes may differentiate into dendritic cells but not into T-lymphocytes. MSCs may interplay with T-lymphocytes modulating their function. This part of introduction need to be rephrased accordingly.

R: We thanks the reviewer for the kind suggestion. The sentence was rephrased accordingly to the referee's comments. Thank you very much.

Lines 114-116  “…to treat in vivo the steroid-resistant graft-versus-host disease (GVHD) in patients underwent either allogeneic organ or hematopoietic stem cells transplantation….”. The GvHD is a systemic disease occurred after hematopoietic stem cells transplantation not after organ transplantation. This statement should be rephrased accordingly.

R: We thanks the reviewer for the kind suggestion. The sentence was rephrased accordingly to the referee's comments. Thank you very much.

This manuscript need professional English editing.

R. The english has been thoroughly revised by an English Language professional. Thank you.

Reviewer 2 Report

The manuscript submitted is focused in a very interesting subject but must be deeply modified. The authors should develop more about the MSCs available for treatment (routes of administration, results already obtained, …), also should describe more precisely the mechanisms of action of these MSCs, also referring previous and recent published work.

At the moment, the manuscript is very poor concerning the information and bibliographic review included.

Author Response

We thank the Reviewers for the considerable attention and the valuable comments that certainly helped us to improve the quality of the present paper. We have revised the manuscript according to the Reviewers’ comments. A revision of the article has been carried out (underlined in yellow for Referee 1 and green for Referee 2, in the main manuscript text).

Please let us know if the revised paper satisfies requirements for publication.

Thank you very much for your attention and courtesy.

Reply to Referee  report II (green in the text):

The manuscript submitted is focused on a very interesting subject but must be deeply modified. The authors should develop more about the MSCs available for treatment (routes of administration, results already obtained, …), also should describe more precisely the mechanisms of action of these MSCs, also referring to previous and recent published work.

R. We really appreciated the comments of reviewer 2 and we proceeded to include within the manuscript the suggested modifications. However, the core of the current scientific hypothesis was based on the children's immunity towards Sars-CoV-2 and not specifically on the use of MSCs and their clinical use (a topic that was in any case implemented as per the reviewer's suggestion). Thank you very much for your understanding.

 At the moment, the manuscript is very poor concerning the information and bibliographic review included.

 R. The bibliography was improved and added as per the reviewer's kind request. Thank you.

Round 2

Reviewer 1 Report

The Authors addressed to my comments sufficiently. 

Reviewer 2 Report

The authors improved the manuscript acording to the suggestions.